# The Effect of Tirzepatide on Body Composition in People with Overweight and Obesity: A Systematic Review of Randomized, Controlled Studies

**DOI:** 10.3390/diseases12090204

**Published:** 2024-09-05

**Authors:** Vincenzo Rochira, Carla Greco, Stefano Boni, Francesco Costantino, Leonardo Dalla Valentina, Eleonora Zanni, Leila Itani, Marwan El Ghoch

**Affiliations:** 1Unit of Endocrinology, Department of Biomedical, Metabolic and Neural Sciences, University of Modena and Reggio Emilia, 41126 Modena, Italy; vincenzo.rochira@unimore.it (V.R.); carlagreco@unimore.it (C.G.); stefano.bno@gmail.com (S.B.); f.costantino96@gmail.com (F.C.); leonardodallavalentina@outlook.it (L.D.V.); eleonora.zanni1995@gmail.com (E.Z.); 2Unit of Endocrinology, Department of Medical Specialties, Azienda Ospedaliero-Universitaria of Modena, 41126 Modena, Italy; 3Department of Nutrition and Dietetics, Faculty of Health Sciences, Beirut Arab University, Riad El Solh, Beirut 11072809, Lebanon; l.itani@bau.edu.lb; 4Center for the Study of Metabolism, Body Composition and Lifestyle, Department of Biomedical, Metabolic and Neural Sciences, University of Modena and Reggio Emilia, 41125 Modena, Italy

**Keywords:** body composition, obesity, randomized controlled trials, systematic review, tirzepatide

## Abstract

Tirzepatide (TZP) is a new anti-obesity drug, and little is currently known about its effect on body composition (BC) in people with overweight or obesity. The aim of this study is to conduct a systematic review on the impact of TZP on BC compartments in this population during weight loss programs. Literature searches, study selection, method development, and quality appraisal were performed. The data were synthesized using a narrative approach, in accordance with the Preferred Reporting Items for Systematic reviews and Meta-Analyses (PRISMA) guidelines. Of the 1379 papers retrieved, 6 randomized controlled trials met the inclusion criteria and were reviewed, revealing the following findings. Firstly, TZP was shown to result in a significant reduction in total fat mass (FM), visceral adipose tissue (VAT) and waist circumference (WC) between baseline and short as well as intermediate follow-ups. Compared to other anti-obesity medications (e.g., dulaglutide and semaglutide) taken over the same duration, TZP showed a superior decrease in body fat compartments (i.e., total FM, VAT and WC). Finally, the effect of TZP on fat-free mass (FFM) is still uncertain because the findings remain inconclusive. In conclusion, TZP appears to be an effective strategy for achieving significant improvements in body fat and its distribution, but additional investigations are still needed to determine the impact of TZP on lean mass in this population.

## 1. Introduction

Obesity is a chronic disease, which represents an increasing global problem, and it has become one of the most serious health conditions worldwide [1]. It is associated with several medical and psychosocial comorbidities that lead to an increase in disability and mortality [2,3]. This has prompted international guidelines to recommend a wide range of weight loss interventions [4,5,6,7]. Several strategies are currently available and used for the management of obesity [2]. The first step and cornerstone approach is weight loss through lifestyle modification programs based on strategies to determine behavioral changes, combined with increased physical activity levels and nutritional recommendations (i.e., low-calorie Mediterranean diets) [8,9]. However, these programs are not always successful, especially in the long term [10]. At the other end of the spectrum, bariatric surgery is a very effective strategy for weight loss [11], but is mostly indicated for individuals with severe obesity (i.e., body mass index (BMI) ≥ 40 kg/m^2^ or ≥35 kg/m^2^ with obesity-associated comorbidities) [12]. In addition, it is not exempt from risks and complications [13], and not always a reversible procedure [14]. Moreover, its availability is limited [15], due to the high cost to healthcare systems [16].

In the middle of this spectrum is the anti-obesity drugs strategy. This field has evolved greatly in the past few years, achieving significant advances and determining a new era of obesity management [17]. In particular, a new drug has been developed called Tirzepatide (TZP), which is a combination of glucagon-like peptide-1 (GLP-1) and a glucose-dependent insulin tropic-polypeptide (GIP) receptor co-agonist [18]. This medication showed a high effectiveness in terms of weight loss outcomes that reached almost a 25% weight loss percentage (WL%) over a period of 1.5 years [18]. In addition, TZP was revealed to have beneficial effects in terms of obesity comorbidities beyond WL [19], such as improvements in type 2 diabetes (T2D) [20], as well as cardiovascular diseases (CVDs) [21], non-alcoholic fatty liver disease (NAFLD) [22] and obstructive sleep apnea syndrome (OSAS) [23]. However, TZP has some adverse effects that are mainly gastrointestinal in nature, and among these, nausea and diarrhea are the most frequently reported [24]. Finally, with regard to drug interactions, while initiating TZP, a dose reduction of concomitantly administered insulin or insulin secretagogues (e.g., sulfonylureas) should be carefully considered to avoid hypoglycemia [25]. Moreover, since TZP is known to delay gastric emptying, this can impact the absorption of concomitantly administered oral medications. Caution is therefore needed in relation to medications, as their efficacy is based on threshold concentrations—in particular, those with a narrow range of therapeutic index (e.g., warfarin) [25]. Similarly, patients on oral contraceptives should be instructed to switch to non-oral contraceptive methods [25].

As per its definition, obesity is characterized by an excessive and abnormal accumulation of body fat, especially in the central regions (visceral adipose tissue, VAT) [26]. In addition, a high proportion of people with obesity (40–50%)—especially those in weight management settings, regardless of their age and gender—are also affected by reduced muscle mass and strength and identified as having the commonly referred phenotype of sarcopenic obesity (SO) [27]. In this context, recent studies have been conducted on the effect of TZP on BC compartments. However, overall its impact in this area is still unclear. If we keep in mind the great magnitude of WL determined by TZP, two clinical considerations should be taken into account: first, it is expected that TZP may have a superior effect in terms of FM reduction in comparison to other obesity medications. Second, at the same time it is legitimately questionable whether TZP may also determine a pronounced reduction in FFM due to a high rate of WL, leading to a deterioration in the above-mentioned condition prevalent in people with obesity or, in other words, SO [28]. To the best of our knowledge, no systematic review posing this issue as a primary outcome (i.e., the effect of TZP on BC compartments) has yet been conducted in order to provide an unbiased interpretation of the evidence published to date. In the light of these considerations, the aim of the current study is to systematically review the published literature to determine the effect of TZP on total and segmental BC compartments according to the following population, intervention, comparison, outcome (PICO) statements [29].

### PICO Statements

We conducted a systematic review on the topic in adherence with the PICO framework [29] as follows:

Population (P): Participants with overweight or obesity, however defined (i.e., body mass index (BMI), waist circumference (WC), fat mass (FM), visceral adipose tissue (VAT), etc.), estimated (anthropometry, skinfold thickness, or bioelectrical impedance analysis (BIA)), or measured (dual-energy x-ray absorptiometry (DXA), air displacement plethysmography (ADP), computed tomography (CT) scan, or magnetic resonance imagining (MRI)), with or without comorbidities.Intervention (I): Short-, intermediate-, or long-term weight loss whether or not followed by a period of weight maintenance by means of anti-obesity or anti-diabetes drugs or other interventions (i.e., lifestyle modification).Comparison (C): Weight loss programs involving TZP as a treatment for obesity/overweight, compared to any other anti-obesity or anti-diabetes drug or placebo defined by the authors (whenever available).Outcome (O): Changes in BC variables directly or indirectly measured and their surrogates as follows:*(i)* Waist circumference (WC)
Mean difference in WC (cm) between baseline and last available follow-up in the TZP group, however expressed.Comparison in changes in WC (cm) between the TZP group vs. any other anti-obesity or anti-diabetes drugs or placebo, however expressed.*(ii)* Fat mass (FM)
Mean difference in FM (kg) between baseline and last available follow-up within the TZP group, however expressed.Comparison in changes in FM (kg) between TZP vs. any other anti-obesity or anti-diabetes drugs or placebo, however expressed.
*(iii)* Visceral adipose tissue (VAT)
Mean difference in VAT (g, cm^2^ or liters) between baseline and last available follow-up within the TZP group, however expressed.Comparison in changes in VAT (g, cm^2^ or liters) in TZP vs. other anti-obesity or anti-diabetes drugs or placebo, however expressed.
*(iv)* Fat-free mass (FFM)
Mean difference in FFM (kg) between baseline and last available follow-up within TZP group, however expressed.Comparison in changes in FFM (kg) between TZP vs. other anti-obesity or anti-diabetes drugs or placebos, however expressed.



## 2. Methods

The systematic review was conducted according to the Preferred Reporting Items for Systematic reviews and Meta-Analyses (PRISMA) guidelines [30] and registered with the International Prospective Register of Systematic Reviews (PROSPERO) as “The Effect of Tirzepatide on Body Composition in People with Overweight and Obesity: A Systematic Review of Randomized Controlled Studies” (CRD42024573477).

### 2.1. Inclusion and Exclusion Criteria

We included all papers that considered TZP as an anti-obesity drug for weight loss intervention and evaluated the changes in BC compartments, however they were expressed, before and after TZP, if they met the following inclusion criteria: (i) publications written in English and (ii) original articles exclusively defined as randomized controlled trials (RCTs). Any other work that was identified with another design, such as a prospective/retrospective observational, experimental, or quasi-experimental non-controlled study, as well as review, cross-sectional, non-controlled, or non-original articles (i.e., case reports, editorials, letters to the editor, and book chapters), was excluded.

### 2.2. Information Sources and Search Strategy

The literature search was performed independently and in duplicate by two authors involved in the conduct of the systematic review. The PubMed and Scopus databases were systematically screened using the following Medical Subject Headings (MeSH) terms and their combinations: #1 Tirzepatide; #2 LY3298176; #3 zepbound; #4 body composition*; #5 lean mass; #6 muscle*; #7 muscle loss; #8 fat mass distribution; #9 fat mass; #10 lean mass loss; #11 appendicular lean mass; #12 appendicular lean mass; #13 sarcopenic obesity; #14 sarcopenia; #15 sarco* #16 dual energy x-ray absorptiometry; #17 DXA; #18 DEXA; #19 bioimped*; #20 BIA; #21 VAT; #22 waist circumference*; and #23 waist circ*; #24 total fat mass; #25 adipose tissue; #26 anthropometric measur*; #27 obesity; #28 type 2 diabetes, according to the following combinations: (((#1) OR (#2) OR (#3)) AND ((#4) OR (#5) OR (#6) OR (#7) OR (#8) OR (#9) OR (#10) OR (#11) OR (#12) OR (#13) OR (#14) OR (#15) OR (#16) OR (#17) OR (#18) OR (#19) OR (#20) OR (#21) OR (#22) OR (#23) OR (#24) OR (#25) OR (#26) OR (#27) OR (#28))). 

In addition, a manual search was carried out to retrieve other articles that had not been identified via the initial search strategy. The publication date was not considered an exclusion criterion for the purposes of this systematic review. 

### 2.3. Study Selection and Quality and Risk-of-Bias Assessments

The two investigators involved in the systematic review independently screened the studies according to their method of conduct and whether they would be appropriate to be included. The quality of each study was determined according to the Jadad scale [31], which relies on three different criteria: randomization (two points), blinding (two points) and a description of dropouts (one point). Out of a total of five points, a score ≥ 3 is indicative that the trial is of good quality [31]. Moreover, the included studies underwent a risk-of-bias assessment to determine whether the standards of reporting for RCTs were satisfied [32] according to the following items: the randomization method, allocation sequence concealment, participant blinding, outcome assessor blinding, outcome measurement, interventionist training and clustering, as well as withdrawals, intent-to-treat analyses, and baseline characteristics. Each study was assigned a “yes” for each criterion it satisfied, and a “no” for each it did not, while “not reported” was used where information for evaluation was insufficient or unavailable. Any disagreement was resolved by consensus discussion between both the reviewers and the principal investigator (PI) in order to take a final decision [33].

### 2.4. Data Collection Process and Data Items and Synthesis

Firstly, both the title and abstract of each paper that had been identified through the electronic or manual search strategies were assessed by two independent authors involved in the conduct of the systematic review for language suitability and topic matter relevance. In a second step, the remaining papers were evaluated for their appropriateness in terms of the inclusion criteria and quality of method. The studies that met the inclusion criteria were presented as a narrative synthesis.

## 3. Results

A total of 1379 papers were retrieved from the initial search, and 355 were immediately eliminated due to being duplicates, with 1024 screened reports remaining. In the first round of screening (titles and abstracts), 996 papers were excluded on the following grounds: the study was (i) written in a language other from English, (ii) was not conducted on humans, (iii) was not identified as an RCT or (iv) was an RCT but did not involve TZP, or people with T2D or obesity. In the second round of the screening of the remaining 28 articles, the full-text papers were assessed for eligibility. Among these, 22 were removed on the following bases: (i) 14 because the participants were on medications other than TZP that might interfere with body weight and BC [20,21,34,35,36,37,38,39,40,41,42,43,44,45]; (ii) two due to the short duration of the TZP treatment [46,47]; (iii) one because it had an open-label lead-in period [48]; and (iv) five for not reporting data related to WC [49,50,51,52,53] (Figure 1).

At the end, six RCTs were available for systematic review and included in the narrative synthesis [22,54,55,56,57,58]. The first author, year of publication, study design, country of conduct, study sample (stratified also by intervention arm), mean baseline BMI and age, follow-up duration, BC assessment method and BC outcomes of each work are reported in Table 1. The Jadad scale checklist indicated that the RCTs (n = 6) were of a high quality (mean score 4.67 points) and had a very low risk of bias (Table 2). 

### Narrative Synthsesis

Yabe et al. [55] conducted the SURPASS J-mono study in Japan, which included 48 participants with T2D that were randomized to receive TZP at 5, 10 or 15 mg or dulaglutide at 0.75 mg once a week, over a follow-up period of 52 weeks. The baseline mean age and BMI were 58.6 ± 7.5 years and 27.5 ± 3.5 kg/m^2^ respectively. After 52 weeks of treatment, the total FM decreased in all TZP groups compared to the baseline (−4.1 kg, *p* < 0.05; −6.8 kg, *p* < 0.001; and −6.6 kg, *p* < 0.001 in the 5, 10 and 15 mg TZP groups, respectively). Moreover, the 10 and 15 mg TZP groups showed a superior effect in the reduction of total FM, exceeding that derived from the dulaglutide (−6.6 kg, *p* < 0.001 and −6.4 kg, *p* < 0.01, respectively). On the other hand, after 52 weeks of treatment, the FFM dropped significantly in the 10 mg and 15 mg TZP groups from the baseline to week 52 (−1.9 kg, *p* < 0.01; and −2.3 kg, *p* < 0.01, respectively), but only in the 15 mg TZP group was it significantly higher than that in the dulaglutide group at week 52 (−2.0 kg, *p* < 0.05) (Table 1).

Jastreboff et al. [58] performed a multicenter, international RCT that included a total of 2539 individuals with overweight or obesity that were randomized to receive TZP (5, 10 or 15 mg) or a placebo once a week, over a follow-up period of 72 weeks. All the participants were assessed for WC, and at week 72, TZP at any dosage resulted in a greater reduction in WC than the placebo: −14.0 (−14.9 to −13.1) cm, −17.7 (−18.7 to −16.8) cm, −18.5 (−19.3 to −17.6) cm and −4.0 (−5.1 to −2.8) cm in the 5 mg, 10 mg, 15 mg TZP and placebo groups, respectively. In the same study, 255 participants also underwent DXA to assess BC, but DXA data were only available for 160 of those patients at 72 weeks of follow-up [58]. The change in total FM from the baseline to week 72 was −33.9% in the TZP groups, compared with −8.2% in the placebo group, with an estimated treatment difference between TZP and the placebo of 25.7%. Furthermore, the total FFM change from the baseline was −10.9% in the TZP groups, in comparison to −2.6% in the placebo group, with an estimated treatment difference between TZP and the placebo of 8.3%. The total FM-to-FFM ratio decreased in the TZP group from 0.93 (baseline) to 0.70 (at week 72) and from 0.95 (baseline) to 0.88 (at week 72) in the placebo group (Table 1).

Gastaldelli et al. [22] conducted a sub-study of the SURPASS-3 trial that included 296 participants with T2D randomized to receive TZP (5, 10, 15 mg) once a week or titrated insulin degludec once per day, with a follow-up period that lasted for 52 weeks. The mean age was 56.2 ± 9.8 years, and the mean BMI was 33.5 ± 4.8 kg/m^2^. All the participants underwent an MRI scan at baseline and at week 52 to assess VAT and abdominal subcutaneous adipose tissue (ASAT). Only 246 out of 296 participants had a post-baseline MRI scan. The VAT decreased significantly with 5 mg (−1.10 ± 0.19 L, *p* < 0.0001), 10 mg (−1.53 ± 0.18 L, *p* < 0.0001) and 15 mg TZP (−1.65 ± 0.18 L, *p* < 0.0001), and significantly increased with insulin degludec (0.38 ± 0.18 L, *p* = 0.04). In addition, ASAT dropped significantly in the TZP groups (−1.4 ± 0.25 L, *p* < 0.0001; −2.25 ± 0.24 L, *p* < 0.0001; and −2.05 ± 0.23 L, *p* < 0.0001 in the 5 mg, 10 mg and 15 mg groups, respectively). A significant increase was found in the insulin degludec group (0.63 ± 0.24 L, *p* = 0.0092). Moreover, the VAT-to-ASAT ratio percentage declined in the TZP groups (−2 ± 0.62%, *p* = 0.0014; −4.96 ± 0.56%, and *p* < 0.0001; −4.36 ± 0.56%, *p* < 0.0001 in the 5 mg, 10 mg and 15 mg groups, respectively), whereas no significant change was observed in the insulin degludec group (1.06 ± 0.57%, *p* = 0.065) (Table 1).

In 2023, Heise et al. [54] conducted a sub-analysis of an RCT on the effect of TZP on BC in patients with T2D, in which 117 participants were randomized to receive 15 mg TZP, 1 mg semaglutide or a placebo once a week, over a follow-up period of 28 weeks. Each patient underwent a plethysmography (using the BOD POD) to assess the FM and the FFM at baseline and after 28 weeks of treatment. Only the participants taking weekly TZP or semaglutide exhibited a significant reduction in total FM (36.8 ± 11.5 kg at baseline and 26.9 ± 0.85 kg at week 28 in the TZP group; 35.3 ± 8.0 kg at baseline and 30.7 ± 0.83 kg at week 28 in the semaglutide group; 38.6 ± 10.7 kg at baseline and 36.6 ± 1.11 kg at week 28 in the placebo group) and FFM (57.7 ± 9.3 kg at baseline and 55.8 ± 0.25 kg at week 28 in the TZP group; 56.3 ± 10.3 kg at baseline and 56.7 ± 0.24 kg at week 28 in the semaglutide group; 59.1 ± 10.3 kg at baseline and 57.3 ± 0.32 kg at week 28 in the placebo group) from the baseline. Furthermore, the FM% loss was significantly higher in the group taking TZP versus those receiving semaglutide (−3.1% [4.9, 1.2]; *p* =0.001), as was the decrease in FFM (−0.8 vs. 1.6 kg; *p* = 0.018) (Table 1).

Wadden et al. [56] conducted an 84-week (12 + 72 weeks) multicenter, international, randomized, parallel-arm, double-blind, placebo-controlled, phase 3 trial to assess the efficacy of TZP in adults with obesity or overweight who successfully lost ≥5% of their baseline weight during a 12-week lead-in period with an intensive lifestyle intervention. Subsequently, 579 participants were then randomized to receive TZP at the maximum tolerated dose (10 or 15 mg) or a placebo once a week. In the TZP group, 86.4% of participants reached a maximum tolerated dose of 15 mg. At week 72, the change in WC in the TZP group was greater than in the placebo one (−14.6 ± 0.7 cm in the TZP group, +0.2 ± 1.0 cm in the placebo group) (Table 1).

In 2024, Zhao et al. [57] conducted an RCT in China known as the SURMOUNT-CN trial among individuals with overweight or obesity. A total of 210 participants were randomly assigned to receive subcutaneous 10 mg or 15 mg TZP or a placebo once a week, plus a lifestyle intervention, for 52 weeks. The mean age was 36.1 ± 9.1 years, and the mean BMI was 32.3 ± 3.8 kg/m^2^. At the end of the study, the participants on TZP exhibited a consistent body weight reduction compared to the placebo group, associated, along the same line, with a more pronounced decrease in WC that was dose-dependent on TZP (difference between TZP 10 mg and the placebo, −8.7 cm [95% CI, −11.2 cm to −6.2 cm; *p* < 0.001]; difference between TZP 15 mg and the placebo, −11.8 cm [95% CI, −14.4 cm to −9.3 cm; *p* < 0.001]) (Table 1).

## 4. Discussion

The aim of this paper was to provide preliminary data on the effects of TZP in terms of changes in BC patterns, as well as to compare its impact with that of other anti-obesity medications on these outcomes. The systematic review included six studies that were objectively judged to be mainly of high quality with a very low risk of bias, and it yielded two main findings.

### 4.1. Main Findings

The first was that TZP is associated with a significant reduction in total FM with respect to the baseline over short (i.e., ≤28 weeks) [54] and intermediate (i.e., 52 and 72 weeks) follow-up periods [55,58]. Moreover, the impact of TZP on the decrease in the total FM seems to exceed that of the other anti-obesity drugs (i.e., dulaglutide and semaglutide) and the placebo (lifestyle modification), but this apparently occurs at higher doses of TZP (i.e., 10 and 15 mg) [54,55,58]. Secondly, TZP at any dose appears to determine a significant decrement in VAT with respect to the baseline over the intermediate follow-up period (i.e., 52 weeks), and its effect seemingly exceeds insulin over the same follow-up duration [22]. To support this finding, data deriving from studies on changes in WC—an anthropometric measure widely accepted as a basic method of assessing body composition, specifically representing a surrogate indirect expression of visceral adiposity (i.e., VAT) [59]—reported a significant reduction exceeding that determined by the placebo over the intermediate follow-up periods (i.e., 52 and 72 weeks) [56,57,58]. Therefore, in the bigger picture, TZP may be considered an effective treatment, which determines a significant drop in FM and can improve its distribution, as its effect also seems to be significantly larger when compared to other anti-obesity or anti-diabetes medications used over a similar follow-up duration.

In this systematic review, we are not in a position to explain the exact mechanism behind the additional beneficial impact of TZP on FM and its distribution, since this is still unclear and not fully understood [60]. However, we can speculate that the dual action of the receptor agonist of GLP-1/GIP of TZP, which can act synergically, may at least in part explain its effect on BC compartments. On a general scale, GLP-1RA reduces fat synthesis and deposition through the regulation of adipose tissue (AT) (lipolysis, fatty acid oxidation and adipocyte differentiation) [61], and this specifically seems to determine a decrease in VAT that is more pronounced in comparison to the reduction in SAT [62]. In addition, GLP-1RA is involved in a relevant process defined as the browning of white adipose tissue (WAT), and it also appears that GLP-1RA activates brown adipose tissue (BAT) [63]. On the other hand, the GIP RA reduces WAT storage expansion, as well as the lipid spillover from the latter to ectopic sites in various organs [64]. This may also explain the greater effect of TZP in reducing BF compared to GLP-1RA alone. This is in line with previous studies that have reported the superiority of TZP with respect to only GLP-1RA on other health outcomes, such as a recent systematic review that showed TZP had a more pronounced effect on HbA1c and weight reduction compared with semaglutide in people with TZP [65].

### 4.2. Clinical Implications

Our systematic review has some clinical implications. Primarily, awareness should be raised among health professionals dealing with obesity and its management through anti-obesity drugs strategies about the effect of TZP on BC compartments from a personalized medicine perspective, especially in terms of body fat and beyond weight loss, and this should be openly discussed with patients.

### 4.3. Strength and Limitations

This study has several strengths. For instance, it is the first systematic review to be conducted on the effect of TZP on BC in patients affected by overweight or obesity with or without T2D and that rigorously adhered to the PRISMA guideline standards; this made its methodology robust, to support the validity of the conclusions related to the findings on body fat compartments. In fact, the studies included in this systematic review were well designed, namely, with suitable randomized samples and appropriate control groups. In other words, the current systematic review was conducted on RCTs that are considered the gold standard in clinical research to assess treatment effectiveness (e.g., medications) [66]. Last but not least, the tools used in the majority of the included studies in this systematic review to assess the BC outcomes (DXA, ADP and MRI) have been widely validated within both clinical and research settings in this population [67].

On the other hand, the findings of this systematic review should be interpreted with caution since it does have some limitations, the foremost being those related to FFM, leading to difficulties in relation to drawing a clear understanding and firm conclusions with regard to the impact of TZP and FFM. In detail, the findings on FFM derived from small-sampled trials, specifically three [54,55,58]. In one study, the findings reported on FFM at different times (i.e., 32 vs. 52 weeks) appear to be inconsistent and do not seem reasonable. In fact, certain methodological limitations can be identified in this study, as the sample size did not exceed 10 participants in each TZP arm, and this may have had an impact on the power of the analysis, as well as the results and conclusions [55]. In addition, in the same study the BIA has been used to measure lean mass (i.e., FFM), while the use of this technique for this aim (i.e., FFM assessment) in people with obesity remains highly debatable [68], especially during weight loss periods, where alterations in hydration status are usually noticed, as reported by the authors themselves in this study [55]. In the second study, despite the fact that the patients on TZP displayed a higher FFM reduction compared to those on semaglutide by only 0.8 kg, the authors considered only the effect of TZP at 15 mg, with no data available on lower TZP doses (i.e., 5 mg and 10 mg), which potentially may have had a more protective effect on lean mass [54]. Similarly, in the last trial, the patients received different doses of TZP, and the analysis was conducted collectively on the entire sample. It was therefore difficult to rule out any potential difference in FFM reduction at different doses [58]. Moreover, on a general scale in our systematic review, we were not able to perform a meta-analysis due to the lack of homogeneity in the BC assessment methods and outcome reporting, which differed widely between the included studies. Finally, the lack of data on adolescents and a longer follow-up duration in adults (i.e., >1–1.5 years) limits the ability to generalize the findings of this systematic review across to the entire population affected by overweight or obesity [69].

### 4.4. New Directions for Future Research

Future works are still necessary on several areas in this topic. Firstly, further investigations that replicate and confirm the findings regarding the beneficial effect of TZP on FM compartments and distribution are vitally needed, since to date the available number of studies is not large, and the majority of them have been conducted in research settings (RCTs), not in real-world clinical ones.

Secondly, regarding FFM, the findings are still inconclusive; therefore, no firm deduction can be made on the matter. In other words, it is still unknown whether TZP may have an adverse or a beneficial impact on FFM during weight loss treatment, from the baseline to follow-up endpoints, or whether TZP can determine a smaller or larger reduction in FFM in comparison to other anti-obesity drugs. On the one hand, TZP seems to determine a higher reduction/deterioration in FFM compared to that of other medications (dulaglutide and semaglutide) by nearly 0.8–2 kg, which appears to occur at the highest dose of TZP (i.e., 15 mg) [54,55]. On the other, TZP apparently causes a greater reduction in fat over lean mass loss by approximately three times [58], which is a similar outcome to what happens during traditional diet-induced weight loss in individuals with obesity, that also accounts for one third (i.e., 20–30%) of FFM [70]. Therefore, the decrease in FFM is an unavoidable phenomenon with TZP, as well as in any other drug or weight loss strategy (i.e., diet, bariatric surgery, etc.). However, we speculate that TZP may have the potential to attenuate the FFM loss in comparison to other anti-obesity drugs due to some novel potential mechanisms that have been recently described in mainly preclinical studies. This includes the impact of GLP-1RA, which appears to increase vascular blood flow and activate glucose delivery into skeletal muscle via the AMP-activated protein kinase (AMPK), thereby promoting muscle synthesis and reducing muscle breakdown [71,72]. In addition, GIP RA was found to suppress intramuscular adipose tissue accumulation [73]. Therefore, it should be expected that TZP could potentially attenuate greater FFM loss and ameliorate sarcopenia. However, this hypothesis should be properly studied in order for it to be confirmed or disproved in clinical settings, due to the importance of this topic. Finally, a better understanding remains necessary of the mechanism of action of TZP, as well as the consequences on health outcomes relating to the BC changes determined by this drug, regardless of the generic effect of weight loss.

## 5. Conclusions

TZP is a new anti-obesity drug, which has been demonstrated to be effective in terms of weight loss, as well as determining significant improvements in obesity comorbidities. After a careful systematic review of the current evidence, TZP can be recommended because of its beneficial impact on BC compartments, specifically through reducing body fat and central obesity. However, future studies are still needed to investigate the impact of TZP on FFM (i.e., muscle and bone mass).

## Figures and Tables

**Figure 1 diseases-12-00204-f001:**
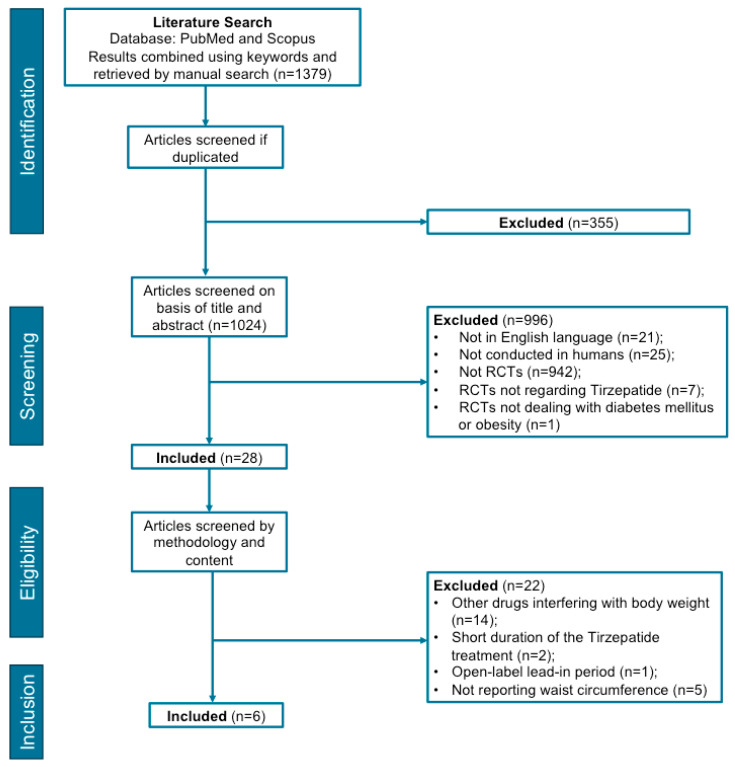
The flowchart summarizing the study selection procedure.

**Table 1 diseases-12-00204-t001:** The RCTs included in the systematic review.

Author and Study Design	Country	Sample	Mean Baseline BMI and Age	Follow-Up Duration	Body Composition Assessment	Body Composition Outcomes ^¥^
Yabe et al., 2023 [55] Multicenter, double-blind RCT	Japan	Total sample: n = 48 (M/F) ➢ By intervention arm: • TZP 5 mg (n = 9) • TZP 10 mg (n = 11) • TZP 15 mg (n = 9) • Dulaglutide 0.75 mg (n = 19)	BMI = 27.5 ± 3.5 kg/m^2^ Age = 58.6 ± 7.5 years	52 weeks	BIA	FM change at week 52 TZP 5 mg: −4.1 (1.7) kg TZP 10 mg: −6.8 (1.5) kg TZP 15 mg: −6.6 (1.6) kg Dulaglutide 0.75 mg: −0.2 (1.1) FFM change at week 52 TZP 5 mg: −1.2 (0.8) kg TZP 10 mg: −1.9 (0.7) kg TZP 15 mg: −2.3 (0.7) kg Dulaglutide 0.75 mg: −0.3 (0.5)
Jastreboff et al., 2022 [58] Multicenter, double-blind RCT	USA, Mexico, Russia, Japan, China, Taiwan, India, South America	Total sample: n = 160 (M/F) ➢ By intervention arm: • Nr	Nr	72 weeks	DXA	FM change at week 72 TZP: −33.9% (pooled 5 mg, 10 mg and 15 mg groups) Placebo: −8.2% FFM change at week 72 TZP: −10.9% (pooled 5 mg, 10 mg and 15 mg groups) Placebo: −2.6% FM/FFM at baseline vs. week 72 TZP 0.93 vs. 0.70 (pooled 5 mg, 10 mg and 15 mg groups) Placebo 0.95 vs. 0.88
*Continued* Jastreboff et al., 2022 [58] Multicenter, double-blind RCT	USA, Mexico, Russia, Japan, China, Taiwan, India, South America	Total sample: n = 2539 (M/F) ➢ By intervention arm: • TZP 5 mg (n = 630) • TZP 10 mg (n = 636) • TZP 15 mg (n = 630) • Placebo (n = 643)	BMI = 38.0 ± 6.8 kg/m^2^ Age = 44.9 ± 12.5 years		WC	Change in WC at week 72 TZP 5 mg: −14.0 (−14.9 to −13.1) cm TZP 10 mg: −17.7 (−18.7 to −16.8) cm TZP 15 mg: −18.5 (−19.3 to −17.6) cm Placebo: −4.0 (−5.1 to −2.8) cm
Gastaldelli et al., 2022 [22] Multicenter, double-blind RCT	USA, Europe, Argentina	Total sample: n= 246 (M/F) ➢ By intervention arm: • TZP 5 mg (n = 71) • TZP 10 mg (n = 79) • TZP 15 mg (n = 72) • Insulin degludec (n = 74)	BMI = 33.5 ± 4.8 kg/m^2^ Age = 56.2 ± 9.8 years	52 weeks	MRI	VAT volume change at week 52 TZP 5 mg: −1.10 L (0.19) TZP 10 mg: −1.53 L (0.18) TZP 15 mg: −1.65 L (0.18) Insulin degludec: +0.38 L (0.18) ASAT volume change at week 52 TZP 5 mg: −1.40 L (0.25) TZP 10 mg: −2.25 L (0.24) TZP 15 mg: −2.05 L (0.23) Insulin degludec: + 0.63 L (0.24)
Heise et al., 2023 [54] Double-blind RCT	Germany	Total sample: n = 117 (M/F) ➢ By intervention arms: • TZP 15 mg (n = 45) • Semaglutide 1 mg (n = 44) • Placebo (n = 28)	Nr	28 weeks	Bod Pod	FM at baseline TZP 15 mg: 36.8 ±11.5 kg Semaglutide 1 mg: 35.3 ± 8.0 kg Placebo: 38.6 ± 10.7 kg FM after 28 weeks TZP 15 mg: 26.9 (0.85) kg Semaglutide 1 mg: 30.7 (0.83) kg Placebo: 36.6 (1.11) kg FFM at baseline TZP 15 mg: 57.7 ± 9.3 kg Semaglutide 1 mg: 56.3 ± 10.3 kg Placebo: 59.1 ± 10.3 kg FFM after 28 weeks TZP 15 mg: 55.8 (0.25) kg Semaglutide 1 mg: 56.7 (0.24) kg Placebo: 57.3 (0.32) kg
Wadden et al., 2023 [56] Multicenter, double-blind RCT	USA, South America	Total sample: n = 579 (M/F) ➢ By intervention arm: • TZP 10 mg (n = 39) • TZP15 mg (n = 248) • Placebo (n = 292)	BMI = 35.9 ± 6.3 kg/m^2^ Age = 45.6 ± 12.2 years	72 weeks	WC	Mean WC absolute change at week 72 TZP: −14.6 (0.7) cm (maximum tolerated dose, 10 or 15 mg) Placebo: +0.2 (1.0) cm
Zhao et al., 2024 [57] Multicenter, double-blind RCT	China	Total sample: n = 210 (M/F) By intervention arm: ➢ TZP 10 mg (n = 70) • TZP 15 mg (n = 71) • Placebo (n = 69)	BMI = 32.3 ± 3.8 kg/m^2^ Age = 36.1 ± 9.1 years	52 weeks	WC	Change in WC at week 52 TZP 10 mg: −11.4 cm (−13.2 to −9.6) TZP 15 mg: −14.5 cm (−16.3 to −12.6) Placebo: −2.6 cm (−4.4 to −0.9)

Abbreviations: USA = United States of America; BIA = bioelectrical impedance analysis; WC = waist circumference; M = males; F = females; SD = standard deviation; BMI = body mass index; MRI = magnetic resonance imaging; DXA = dual-energy X-ray absorptiometry; Nr = not reported; FM = fat mass; FFM = fat-free mass; VAT = visceral adipose tissue, ASAT = abdominal subcutaneous adipose tissue, TZP = Tirzepatide. ^¥^ Values are expressed as least squared mean (LSM) standard error (SE) or LSM (95% CI) or mean ± SD.

**Table 2 diseases-12-00204-t002:** Risk-of-bias criteria and Jadad scale.

Author	Yabe et al., 2022 [55]	Jastreboff et al., 2022 [58]	Gastaldelli et al., 2022 [22]	Wadden et al., 2023 [56]	Heise et al., 2023 [54]	Zhao et al., 2024 [57]
Risk of bias						
Was the method of randomization to groups appropriate?	+	+	+	+	+	+
Was the allocation sequence concealed from those assigning patients to groups?	+	+	−	+	+	+
Was the outcome measurement performed in the same manner with similar intensity in all groups being compared?	+	+	+	+	+	+
Were similarly trained individuals administering the intervention across groups?	+	+	+	+	+	+
Were all the withdrawals described?	+	+	+	+	+	+
Were all originally randomized participants analyzed in the groups they were assigned to (i.e., an intention-to-treat analysis)?	+	+	+	+	+	+
Was clustering at the group level accounted for in the analyses? Were the groups similar at baseline?	+	+	+	+	+	+
Jadad Scale						
Randomization	2	2	2	2	2	2
Blinding	2	2	0	2	2	2
Account of all patients	1	1	1	1	1	1
Total Score	5	5	3	5	5	5

Risk-of-bias reporting: Yes: +; no: −; not reported: nr. Jadad scale reporting randomized controlled trials. It evaluates three different items: randomization (0–2), blinding (0–2), and account of all patients (0–1); studies with scores > 3 were considered as good quality.

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
