# Peer review of "The Effect of Tirzepatide on Body Composition in People with Overweight and Obesity: A Systematic Review of Randomized, Controlled Studies"

_diseases, 2024, doi:10.3390/diseases12090204_

Round 1

Reviewer 1 Report

Comments and Suggestions for Authors

check the attached file please

Author Response

Please check the file in attachment. 

Reviewer 2 Report

Comments and Suggestions for Authors

Thanks for this review article

Comments:

1-The number and variety of reviewed articles should be increased, especially the articles that ultimately did not achieve the desired result in the use of TZP.

2-In addition to the beneficial effects of TZP, its side effects should be explained more.

3- Review more articles about TZP drug interactions.

Author Response

Reviewer #2

1- The number and variety of reviewed articles should be increased, especially the articles that ultimately did not achieve the desired result in the use of TZP. Response: we thank the reviewer for the valuable comment. The aim of our systematic was to assess the impact of TZP on body composition, and we have included all the articles related to this topic.

2- In addition to the beneficial effects of TZP, its side effects should be explained more. Response: now we reported the side effects of TZP as suggested, and added a suitable reference (See Page 2, Paragraph 2).

3- Review more articles about TZP drug interactions. Response: now we reported the TZP drug interactions as suggested, and added a suitable reference (See Page 2, Paragraph 2).

Reviewer 3 Report

Comments and Suggestions for Authors

In this manuscript, Dr. V. Rochira and colleagues present a systematic review of the data on the effect of terzipatide on body composition in obese individuals. This is an interesting and relevant topic. The study was performed methodologically correctly, taking into account the PRISMA guidelines. The quality of the analyzed studies was assessed. The results have undoubted clinical significance. In my opinion, the article will be of interest to both researchers and physicians. At the same time, some aspects of the study need clarification or explanation.

1. The aim of this study was to analyze changes in body composition with terzipatide treatment. However, only four of the six studies included in the analysis examined body composition, while the other two studies only measured waist circumference. Waist circumference can hardly be considered a reliable measure of body composition. Please explain why these studies were included in the analysis.

2. Abstract. “Compared to other anti-obesity medications taken over the same duration, TZP showed a superior decrease in body fat compartments (i.e., total FM, VAT and WC). This conclusion appears to be an overgeneralization. Please elaborate or remove it.

3. The Discussion should more clearly highlight the significant heterogeneity in body composition assessment methods and populations across the studies.

4. I would recommend that the Discussion include the fact that in three studies all participants had diabetes. To judge the effect of terzipatide in this group, a brief description of the patients from this point of view (HbA1c, other antihyperglycemic drugs) would be appropriate.

5. The greater effectiveness of terzipatide in reducing fat mass compared to other GLP-1 agonists deserves an explanation.

6. Please, remove a duplicated word in line 45.

7. Table 1. It is not clear how the data on degludec are expressed: insulin degludec + 0.63L (0.24).

Author Response

Reviewer #3

In this manuscript, Dr. V. Rochira and colleagues present a systematic review of the data on the effect of terzipatide on body composition in obese individuals. This is an interesting and relevant topic. The study was performed methodologically correctly, taking into account the PRISMA guidelines. The quality of the analyzed studies was assessed. The results have undoubted clinical significance. In my opinion, the article will be of interest to both researchers and physicians. At the same time, some aspects of the study need clarification or explanation. Response: we thank the reviewer for her/his time spent to review our manuscript as well as for the appreciation. We did our best to reply all the comments.

  1. The aim of this study was to analyze changes in body composition with terzipatide treatment. However, only four of the six studies included in the analysis examined body composition, while the other two studies only measured waist circumference. Waist circumference can hardly be considered a reliable measure of body composition. Please explain why these studies were included in the analysis. Response: waist circumference is an anthropometric measure widely accepted as basic method of assessing body composition, specifically representing a surrogate indirect expression of visceral adiposity (i.e., VAT). This has been clearly mentioned in the text with the suitable reference (See Page 13, Paragraph 2).

  1. Abstract. “Compared to other anti-obesity medications taken over the same duration, TZP showed a superior decrease in body fat compartments (i.e., total FM, VAT and WC). This conclusion appears to be an overgeneralization. Please elaborate or remove it. Response: now we added the names of these medications to be more specific (See Page 1, Abstract).

  1. The Discussion should more clearly highlight the significant heterogeneity in body composition assessment methods and populations across the studies. Response: now this is clearly highlighted in the Discussion section (See Page 14, Paragraph 3).

  1. I would recommend that the Discussion include the fact that in three studies all participants had diabetes. To judge the effect of terzipatide in this group, a brief description of the patients from this point of view (HbA1c, other antihyperglycemic drugs) would be appropriate. Response: in line with the comment of the reviewer now we added a statement and a suitable reference (See Page 13, Paragraph 3).

  1. The greater effectiveness of terzipatide in reducing fat mass compared to other GLP-1 agonists deserves an explanation. Response: This has been explained in the Discussion section in the subsection of Main findings (See Page 13, Paragraph 3).

  1. Please, remove a duplicated word in line 45. Response: the duplicate has been removed as suggested (See Page 2, Paragraph 2).

  1. Table 1. It is not clear how the data on degludec are expressed: insulin degludec + 0.63L (0.24). Response: we missed the colon, now it is expressed correctly as: insulin degludec: + 0.63L (0.24), indicating an increase in VAT, and what is between the parenthesis is the standard error, which was clearly stated in the legend of Table 1.

Round 2

Reviewer 1 Report

Comments and Suggestions for Authors

no further comments

Reviewer 3 Report

Comments and Suggestions for Authors

The manuscript is sufficiently revised.